# Association between higher urinary normetanephrine and insulin resistance in a Japanese population

**Masaya Murabayashi**[1], **Makoto Daimon**[1]*, **Hiroshi Murakami**[1], **Tomoyuki Fujita**[1],
**Eri Sato**[1], **Jutaro Tanabe**[1], **Yuki Matsuhashi**[1], **Shinobu Takayasu**[1], **Miyuki Yanagimachi**[1],
**Ken Terui**[1], **Kazunori Kageyama**[1], **Itoyo Tokuda**[2], **Kaori Sawada**[2], **Kazushige Ihara**[2]

**1** Department of Endocrinology and Metabolism, Hirosaki University Graduate School of Medicine, Hirosaki, Aomori, Japan, **2** Department of Social Medicine, Hirosaki University Graduate School of Medicine, Hirosaki, Aomori, Japan

* mdaimon@hirosaki-u.ac.jp

## Abstract

Since activation of the sympathetic nervous system is associated with both impaired insulin secretion and insulin resistance, or namely with diabetes, evaluation of such activation in ordinary clinical settings may be important. Therefore, we evaluated the relationships between urinary concentrations of the catecholamine metabolites, urinary normetanephrine (U-NM) and urinary metanephrine (U-M), and glucose metabolism in a general population. From 1,148 participants in the 2016 population-based Iwaki study of Japanese, enrolled were 733 individuals (gender (M/F): 320/413; age: 52.1±15.1), who were not on medication affecting serum catecholamines, not diabetic, and had complete data-set and blood glucose levels appropriate for the evaluation of insulin secretion and resistance, using homeostasis model assessment (HOMA-β and HOMA-R, respectively). Univariate linear regression analyses revealed significant correlations between both U-NM and U-M, and HOMA-β, but adjustment for multiple factors correlated with HOMA indices abolished these (β = -0.031, p = 0.499, and β = -0.055, p = 0.135, respectively). However, the correlation between U-NM and HOMA-R observed using univariate linear regression analysis (β = 0.132, p<0.001) remained significant even after these adjustments (β = 0.107, p = 0.007), whereas U-M did not correlate with HOMA-R. Furthermore, use of the optimal cut-off value of U-NM for the prediction of insulin resistance (HOMA-R >1.6) determined by ROC analysis (0.2577 mg/gCr) showed that individuals at risk had an odds ratio of 2.65 (confidence interval: 1.42–4.97) after adjustment for the same factors used above. Higher U-NM concentrations within the physiologic range are a significant risk factor for increased insulin resistance in a general Japanese population.

## Introduction

Type 2 diabetes (DM) is a heterogeneous disorder of glucose metabolism characterized by both insulin resistance and pancreatic β-cell dysfunction. Catecholamines (CAs) are known to

---

**Data Availability Statement:** All relevant data generated or analyzed during this study are included in the manuscript.

---

**Funding:** The Japan Science and Technology Agency.

**Competing interests:** The authors have declared that no competing interests exist.

be among the factors involved in the pathophysiology of DM, as shown by the study of pathologic conditions, such as pheochromocytoma.[1–4] In cases of pheochromocytoma, glucose intolerance is often present,[1–4] and an excess of CAs appears to be the cause.[1, 5–10] However, the mechanisms leading to the glucose intolerance that characterizes pheochromocytoma have not been fully determined. Hyperinsulinemic-euglycemic clamp studies have shown that insulin resistance underpins the glucose intolerance present in patients with pheochromocytoma,[9,11] but another recent study, which evaluated the effects of surgical removal of pheochromocytomas on glucose metabolism showed that impaired insulin secretion is a primary cause of the associated impairment in glucose tolerance.[10] In addition, although CAs have been shown to inhibit insulin secretion *via* α2 receptors on pancreatic β-cell,[12–15] they have also been shown to increase insulin resistance through α1 and the β receptors.[8,9] Furthermore, adrenaline has a higher affinity for α2 receptors than noradrenaline.[2,16] Therefore, the mechanisms whereby CAs reduce insulin secretion and increase insulin resistance differ, and adrenaline and noradrenaline (two principal CAs), may affect glucose metabolism differently.

Differing associations of the urinary concentrations of metanephrine (U-M) and normetanephrine (U-NM), metabolites of adrenaline and noradrenaline, respectively, with glucose intolerance have recently been reported. [17] In this study, which evaluated the relationship between changes in homeostatic model assessment (HOMA) indices and changes in U-M and U-UM concentrations resulting from the surgical removal of pheochromocytomas, the improvement in U-M concentrations was positively associated with the improvement in HOMA-β, an index representing insulin secretion; while the improvement of U-NM concentrations was positively associated with the improvement in HOMA-R, an index representing insulin resistance. Therefore, U-M and U-NM concentrations may reflect different risks for diabetes, at least in the presence of pathologic conditions, such as pheochromocytoma.

However, the relationships between the concentrations of these metabolites and glucose metabolism have not been assessed under physiologic conditions. Namely, it is unknown whether U-M and U-NM concentrations are associated with insulin secretion and/or insulin resistance under physiologic conditions, and how. Therefore, to address these questions, we aimed to evaluate the relationships of U-M and U-MN levels within the physiological range with glucose metabolism in a general population.

## Materials and methods

### Participants

Participants were recruited from the Iwaki study, a health promotion study of Japanese people of over 20 years of age that aims to prevent lifestyle-related diseases and prolong lifespan. The study is conducted annually in the Iwaki area of the city of Hirosaki in Aomori Prefecture, northern Japan.[18,19] Of the 1,148 individuals who participated in the Iwaki study in 2016, the following individuals were excluded from the present study: 330 who were taking drugs that affect serum catecholamine concentrations (i.e. α- and/or β– blockers), five with incomplete clinical data, 78 with diabetes, and two with fasting blood glucose levels < 63 mg/dl or >140 mg/dl to better evaluate HOMA indices. After these exclusions, 733 individuals (320 men, 413 women) aged 52.1 ± 15.1 years were included in our study.

This study was approved by the Ethics Committee of the Hirosaki University School of Medicine (No. 2016–028 (approved at may 27, 2016)), and was conducted in accordance with the recommendations of the Declaration of Helsinki. Written informed consent was obtained from all the participants.

## Measurements made

Blood samples were collected in the morning from a peripheral vein under fasting conditions, while participants were in a supine position. All laboratory testing was performed in a commercial laboratory (LSI Medience Co., Tokyo, Japan), in accordance with the instructions of the vendors. U-M and U-NM concentrations were determined using liquid chromatography-tandem mass spectrometry (LC-MS/MS). The following clinical characteristics were also measured: height, body weight, body mass index, fasting blood glucose, fasting serum insulin, glycated hemoglobin (HbA1c), systolic blood pressure, diastolic blood pressure, and serum total cholesterol, triglyceride, high-density lipoprotein-cholesterol, uric acid, urea nitrogen, and creatinine. HbA1c (%) is expressed using the National Glycohemoglobin Standardization Program value. Insulin resistance and secretion were assessed by homeostasis model assessments, using fasting blood glucose and insulin concentrations (HOMA-R and HOMA-β). Diabetes was defined according to the 2010 Japan Diabetes Society criterion (fasting blood glucose levels ≥ 126 mg/dL).[20] In subjects whose fasting blood glucose concentrations were not measured, diabetes was defined by an HbA1c concentrations ≥ 6.5%. Those taking medication for diabetes were also defined as having diabetes. Hypertension was defined by a blood pressure ≥ 140/90 mmHg or the use of anti-hypertensive therapy. Dyslipidemia was defined by a LDL cholesterol of ≥ 120 mg/dL, an HDL cholesterol of < 40 mg/dL, a triglyceride of ≥ 150 mg/dL, or the use of anti-hyperlipidemic therapy. Alcohol intake status (current or non-drinker) and smoking habits (never, past, or current) were determined using a questionnaire.

## Statistical methods

Clinical characteristics are summarized using means ± SD. The statistical significance of difference between two groups (parametric) and case-control associations between groups (non-parametric) were assessed using analysis of variance (ANOVA) and the $\chi^2$ test, respectively. Correlations between HOMA indices and clinical characteristics, including U-M and U-NM concentrations, were assessed using linear regression analysis. The relationships of U-M and U-NM concentrations with insulin secretion and insulin resistance were evaluated using multiple logistic regression analysis, with adjustment for factors found to be associated with the indices using univariate regression analysis. The relationship between U-NM concentrations and insulin resistance, defined on the basis of HOMA-R (≥1.6), was calculated using multiple logistic regression analysis with adjustment for factors found to be associated with insulin resistance using univariate regression analysis. Receiver operating characteristic (ROC) curve analysis was performed to determine the cut-off value for U-NM that would predict increased insulin resistance. For statistical analyses, HOMA indices, serum creatinine, and U-M and U-NM, were $\log_{10}$-transformed to approximate a normal distribution. $P < 0.05$ was accepted as representing statistical significance. All analyses were performed using JMP Pro version 14.0 (SAS Institute Japan Ltd., Tokyo, Japan).

## Results

### Clinical characteristics of the study subjects

The clinical characteristics of the participants, classified according to sex, are shown in Table 1. Their mean ages were 50.2 ± 15.2 years for men and 53.5 ± 14.9 years for women. Most clinical characteristics significantly differ between men and women, and urinary NM and M concentrations (mg/g creatinine) were significantly lower in men than women (0.16 ± 0.06 *vs.* 0.22 ± 0.09, and 0.11 ± 0.04 *vs.* 0.12 ± 0.05, respectively).

## Relationships of urinary NM and M concentrations with HOMA indices

The univariate correlations between the clinical characteristics and HOMA indices (R and β) are shown in Table 2. Because many clinical characteristics such as age, sex, BMI, body fat percentage, blood pressure, HbA1c, and serum lipid, uric acid, and urea nitrogen concentrations were found to be correlated with HOMA indices, these factors were used as covariates for the adjustment of further analyses. Although univariate regression analyses revealed significant correlations between both U-NM and U-M and HOMA-β (β = -0.154 p<0.001, and β = -0.174, p<0.001, respectively), adjustment for the variables that correlated with HOMA indices abolish these correlations (β = -0.031, p = 0.499, and β = -0.055, p = 0.135, respectively) (Table 3). In contrast, the correlation between U-NM and HOMA-R identified in the univariate regression analyses (β = 0.132, p<0.001) remained significant even after these adjustments (β = 0.107, p = 0.007), whereas U-M did not correlate with HOMA-R (univariate: β = −0.067 p = 0.068; multivariate: β = 0.037, p = 0.276) (Table 4).

## Association between high physiologic U-NM concentrations and increased insulin resistance

To further evaluate the relationship between U-NM and insulin resistance, the participants were allocated to two groups on the basis of their U-NM concentrations (upper: ≥0.18 mg/g

**Table 1. Clinical characteristics of the participants, classified according to sex.**

| Characteristics | Men | Women | p |
|---|---|---|---|
| Number | 320 | 413 | - |
| Age (yr) | 50.2±15.2 | 53.5±14.9 | 0.004** |
| Height (cm) | 169.2±6.7 | 155.8±6.3 | <0.001** |
| Body weight (kg) | 67.6±10.0 | 53.7±8.4 | <0.001** |
| Body mass index (kg/m2) | 23.6±3.0 | 22.1±3.2 | <0.001** |
| Fat (%) | 19.9±5.6 | 29.8±6.8 | <0.001** |
| Urinary NM (mg/gCr) | 0.16±0.06 | 0.22±0.09 | <0.001** |
| Urinary M (mg/gCr) | 0.11±0.04 | 0.12±0.05 | 0.001** |
| Fasting plasma glucose (mg/dl) | 89.9±9.9 | 86.9±9.7 | <0.001** |
| HbA1c (%) | 5.71±0.33 | 5.73±0.31 | 0.38 |
| Fasting serum insulin: IRI (μU/ml) | 5.00±2.69 | 5.11±2.66 | 0.59 |
| HOMA-R | 1.13±0.65 | 1.12±0.68 | 0.92 |
| HOMA-β | 76.7±71.8 | 84.7±45.2 | 0.066 |
| Systolic blood pressure (mmHg) | 124.4±16.5 | 122.2±18.5 | 0.097 |
| Diastolic blood pressure (mmHg) | 77.0±12.1 | 73.3±12.2 | <0.001** |
| LDL cholesterol (mg/dl) | 116.1±27.0 | 117.8±29.5 | 0.423 |
| Triglyceride (mg/dl) | 119.7±77.8 | 80.2±42.7 | <0.001** |
| HDL cholesterol (mg/dl) | 58.5±17.3 | 69.5±16.5 | <0.001** |
| Serum albumin (g/dl) | 4.57±0.29 | 4.45±0.29 | <0.001** |
| Serum uric Acid (mg/dl) | 6.09±1.22 | 4.42±1.00 | <0.001** |
| Serum urea Nitrogen (mg/dl) | 14.6±4.1 | 13.9±4.1 | 0.012* |
| Serum creatinin (mg/dl) | 0.84±0.17 | 0.63±0.12 | <0.001** |
| Hypertension: n (%) | 118(36.9) | 128(31.0) | 0.098 |
| Dyslipidemia: n (%) | 139(43.4) | 171(41.4) | 0.60 |
| Drinking alcohol: n (%) | 233(72.8) | 130(31.5) | <0.001** |
| Smoking (Never/ Past/ Current):n | 125/88/107 | 313/54/46 | <0.001** |

P<0.05 and <0.01 are indicated by * and **, respectively. Data are mean±SD or number of subjects (%).

**Table 2. Factors correlated with HOMA indices.**

| Characteristics | R | | β | |
|---|---|---|---|---|
| | β | p | β | p |
| Sex (F/M) | 0.004 | 0.921 | 0.165 | <0.001** |
| Age (yr) | 0.163 | <0.001** | -0.301 | <0.001** |
| Height (cm) | -0.066 | 0.074 | -0.013 | 0.721 |
| Body weight (kg) | 0.344 | <0.001** | 0.116 | 0.002* |
| Body mass index (kg/m²) | 0.507 | <0.001** | 0.165 | <0.001** |
| Fat (%) | 0.404 | <0.001** | 0.235 | <0.001** |
| Urinary NM (mg/gCr) | 0.132 | <0.001** | -0.154 | <0.001** |
| Urinary M (mg/gCr) | -0.067 | 0.068 | -0.174 | <0.001** |
| HbA1c (%) | 0.306 | <0.001** | -0.198 | <0.001** |
| Systolic blood pressure (mmHg) | 0.279 | <0.001** | -0.132 | <0.001** |
| Diastolic blood pressure (mmHg) | 0.220 | <0.001** | -0.081 | 0.029* |
| LDL cholesterol (mg/dl) | 0.111 | 0.003* | -0.068 | 0.066 |
| Triglyceride (mg/dl) | 0.273 | <0.001** | 0.105 | 0.005** |
| HDL cholesterol (mg/dl) | -0.244 | <0.001** | -0.110 | 0.003** |
| Serum albumin (g/dl) | 0.023 | 0.540 | 0.102 | 0.006** |
| Serum uric Acid (mg/dl) | 0.164 | <0.001** | -0.061 | 0.099 |
| Serum urea Nitrogen (mg/dl) | 0.093 | 0.011* | -0.165 | <0.001** |
| Serum creatinin (mg/dl) | 0.024 | 0.513 | -0.055 | 0.135 |
| Hypertension: n (%) | 0.272 | <0.001** | -0.118 | 0.001* |
| Dyslipidemia: n (%) | 0.251 | <0.001** | 0.023 | 0.538 |
| Drinking alcohol: n (%) | -0.107 | 0.004** | -0.144 | <0.001 |
| Smoking (Never/ Past/ Current):n | -0.138 | <0.001* | -0.029 | 0.434 |

P<0.05 and <0.01 are indicated by * and **, respectively. Data are mean±SD or number of subjects (%).

creatinine, lower: <0.18 mg/g creatinine), because the relationship between U-NM and HOMA-R appeared to be J-shaped, with an inflection point in the mid-range of the U-NM levels (Fig 1). The correlation between U-NM and HOMA-R was stronger in the upper group (univariate: β = 0.197, p<0.001; multivariate: β = 0.152, p = 0.002), but abolished in the lower group (univariate: β = −0.022 p = 0.667; multivariate: β = -0.018, p = 0.714). Then, using the upper group alone, we then evaluated the risk of increased insulin resistance (defined as HOMA-R ≥1.6) according to U-NM concentration, and determined the optimal cut-off value of U-NM for the prediction of increased insulin resistance using ROC analyses (area under the curve: 0.608; sensitivity: 0.558; specificity: 0.662). Using the optimal cut-off value of U-NM

**Table 3. Correlations of urinary NM and M concentrations with HOMA indices.**

| | R[#1] | | | | β[#2] | | | |
|---|---|---|---|---|---|---|---|---|
| | Univariate | | Multiple factors adjusted[#1] | | Univariate | | Multiple factors adjusted[#2] | |
| | β | p | β | p | β | p | β | p |
| Urinary NM | 0.132 | <0.001** | 0.107 | 0.007** | -0.154 | <0.001** | -0.031 | 0.499 |
| Urinary M | -0.067 | 0.068 | 0.037 | 0.276 | -0.174 | <0.001** | -0.055 | 0.135 |

#1: Adjusted for age, body mass index, glycated hemoglobin (HbA1c), serum triglyceride, uric acid, and urea nitrogen, hypertension, alcohol drinking, and smoking.
#2: Adjusted for age, sex, body fat percentage, HbA1c, systolic blood pressure, serum high-density lipoprotein-cholesterol, albumin, and urea nitrogen, and alcohol drinking. P<0.05 and <0.01 are indicated by * and **, respectively.

**Table 4. Risk of insulin resistance associated with urinary NM concentration.**

| | Univariate | | | Multiple factors adjusted | | |
|---|---|---|---|---|---|---|
| | OR | 95%CI | p | OR | 95%CI | p |
| Upper alone | 2.48 | 1.48–4.14 | <0.001** | 2.65 | 1.42–4.95 | 0.002** |
| Whole | 2.80 | 1.82–4.29 | <0.001** | 2.73 | 1.34–5.57 | 0.006** |

Odds ratios (ORs) with 95% confidence intervals (CIs) and p values are shown. Multiple factors were used to adjust the analyses: age, BMI, HbA1c, serum triglyceride, uric acid, and urea nitrogen, hypertension, alcohol drinking, and smoking. $P < 0.05$ and $< 0.01$ are indicated by * and **, respectively.

concentration (0.2577 mg/g creatinine), those at risk had an odds ratio (OR) of 2.65 (p = 0.002, confidence interval (CI): 1.42–4.95) after adjustment for the variables described above. Moreover, even when the all the samples were analyzed, participants with U-NM concentration above the cut-off value were significantly at risk for increased insulin resistance after adjustment for the confounding factors (OR: 2.73, p = 0.006, 95%CI: 1.34–5.57).

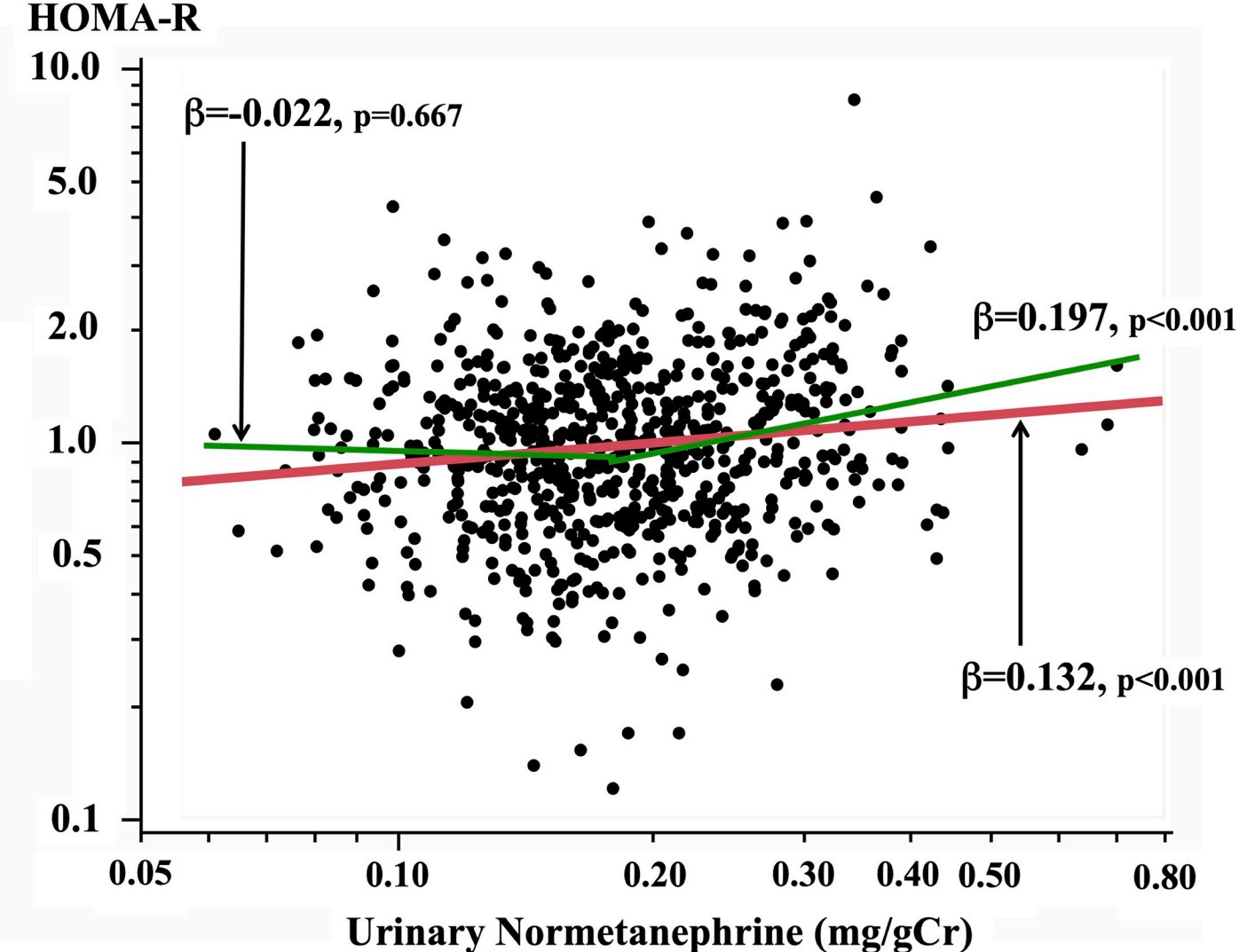

**Fig 1. Correlation between urinary normetanephrine concentration and insulin resistance, assessed using the homeostasis model (HOMA-R).** Linear regression lines are shown for the entire cohort (red line) and halves of the cohort, divided on the basis of their urinary normetanephrine concentration (upper: > 0.18 mg/g creatinine; lower: < 0.18 mg/g creatinine) (green line).

Taken together, these results indicate that higher U-NM concentrations are significantly associated with increased insulin resistance, but not decreased insulin secretion, in a general Japanese population.

## Discussion

In this cross-sectional study of a general Japanese population, we found that U-NM concentrations within the physiologic range significantly correlate with HOMA-R, but not with HOMA-β. The associations between CA concentrations, including those of U-NM and U-M, and the indices, reflecting glucose metabolisms such as HOMA indices, have frequently been reported in patients with pheochromocytomas.[1–10,12] In these patients, CA concentrations has been shown to be positively associated with both insulin resistance and impairment of insulin secretion, both of which have been shown to be primary causes of the impaired glucose tolerance in such patients.[9,10] However, these relationship had not been well studied in individuals with physiologic CA concentrations or healthy individuals. The results observed here indicate that higher U-NM concentration is a risk for insulin resistance, but not for insulin secretion, when the concentrations are within the physiologic range.

As described previously, the type of CA present in excess determined how glucose metabolism is impacted, because the mechanisms whereby CA leading to decreased insulin secretion and increased insulin resistance differ,[8,9,12–15] and adrenaline has higher affinity for α2 receptors on pancreatic β-cell than noradrenaline.[1,16] A previous study of patients with pheochromocytoma found that U-NM and U-M concentrations are differently associated with insulin resistance and insulin secretion: U-M is positively associated with impairment in insulin secretion, but not with insulin resistance, while U-NM is negatively associated with impairment of insulin secretion and positively associated with insulin resistance.[17] In the current study of people with CA concentrations within the physiologic range, the relationship between CAs and HOMA indices also differed according to the type of CA, with U-NM concentrations being significantly associated with insulin resistance, but no other association being identified. Taken together, these findings suggest that noradrenaline, but not adrenaline, excess causes insulin resistance at concentrations within the physiologic range, and when their concentrations reach the pathologic range, adrenaline has the most significant negative effect on insulin secretion.

Because this was a cross-sectional study, we cannot infer a cause and effect relationship, so the positive correlation between U-NM and HOMA-R does not necessarily imply that NM increases insulin resistance. We believe that plasma noradrenalin concentrations within the physiologic range do not have substantial metabolic effects, but rather reflect sympathetic nerve activity. Systemic concentrations of CAs, particularly adrenaline and noradrenaline, appear to represent adrenal function and sympathetic nerve activity, respectively, especially under physiologic conditions.[21] However, because plasma noradrenaline has been shown to be only a small fraction of the quantity secreted from sympathetic nerve terminals,[22, 23] plasma noradrenaline concentration is thought to be a less sensitive indicator of overall sympathetic activity.[23] Here, we used U-M and U-NM concentrations to represent plasma CA concentrations, because these concentrations are known to be stable.[16] Nevertheless, U-MN may, at least in part, reflect sympathetic nerve activity, because the association between high sympathetic activity and insulin resistance has been well documented.[21, 24, 25]

Our study has both strengths and limitations. Significant strengths are that statistical adjustments were made for multiple factors that could confound the results, and a relatively large sample of the general population was studied, which allowed us to evaluate the relationships between CAs and insulin secretion without any influence of compensatory increases in insulin

secretion. Contrary, adjustment for multiple factors can reduce statistical power, and, thus, may also be a limitation. However, statistical power to determine the difference in the frequencies of insulin resistance between the subjects above and below the cut-off value for U-NM determined using SampSize software (http://sampsize.sourceforge.net/iface/index.html) was 99.8% to detect an OR of 2.73 for insulin resistance at a significance level of 0.05, and, thus, the issue does not seem to be substantial. Furthermore, we excluded individuals taking medication that could have affected serum CA concentrations, and those with fasting blood glucose concentrations < 63 mg/dl or > 140 mg/dl, to better evaluate HOMA indices. Such exclusions made the sample suitable for interrogation of the relationships between CAs and HOMA indices.

Several limitations should be mentioned. Firstly, the participants had enrolled in a health promotion study, rather than a routine health check study, and may therefore have been more invested in keeping themselves healthy than the wider general population. Secondly, 415 of the original 1,148 participants were excluded in the present study, which may have led to selection bias. Taken together, these factors may imply that the participants did not accurately represent the general population. Thirdly, we used HOMA indices (R and β) as surrogates for insulin resistance and secretion, respectively. However, because both indices are calculated using fasting concentrations of insulin and glucose, they are closely related. Thus, when insulin resistance increases, insulin secretion increases in compensation, meaning that both HOMA-R and β increase. Therefore, any associations with HOMAs should be interpreted cautiously, as in the case of the study reporting associations of U-NM with both insulin resistance and insulin secretion, although the association between U-NM and insulin resistance seemed to be primary.[17] However, in the present study, we did not observe such conflicting results. Fourthly, we evaluated insulin secretion using HOMA-β to represent insulin secretion in the fasting state, and did not measure any indices of glucose-stimulated insulin secretion. This is a limitation of the study, because suppression of the acute insulin secretory response has been reported to be the principal effect of adrenaline.[10, 15, 26] Finally, because the study was cross-sectional, rather than a cohort study, we could not determine whether higher U-NM can predict the risk of future glucose intolerance or diabetes.

In conclusion, higher U-NM levels within the physiologic range are significantly associated with insulin resistance, but not with insulin secretion, in a general Japanese population. These results suggest that higher U-NM levels are a risk factor for insulin resistance, and thus for future diabetes. This possibility should be tested in future prospective studies.

## Author Contributions

**Conceptualization:** Masaya Murabayashi, Makoto Daimon.

**Data curation:** Itoyo Tokuda, Kaori Sawada, Kazushige Ihara.

**Formal analysis:** Masaya Murabayashi.

**Funding acquisition:** Makoto Daimon, Kazushige Ihara.

**Investigation:** Masaya Murabayashi, Makoto Daimon, Kazushige Ihara.

**Methodology:** Masaya Murabayashi, Makoto Daimon.

**Project administration:** Ken Terui.

**Resources:** Hiroshi Murakami, Tomoyuki Fujita, Eri Sato, Jutaro Tanabe, Yuki Matsuhashi, Shinobu Takayasu, Miyuki Yanagimachi, Kazunori Kageyama, Itoyo Tokuda, Kaori Sawada, Kazushige Ihara.

**Supervision:** Makoto Daimon, Hiroshi Murakami.

**Validation:** Hiroshi Murakami.

**Writing – original draft:** Makoto Daimon.

**Writing – review & editing:** Masaya Murabayashi, Hiroshi Murakami, Tomoyuki Fujita, Eri Sato, Jutaro Tanabe, Yuki Matsuhashi, Shinobu Takayasu, Miyuki Yanagimachi, Ken Terui, Kazunori Kageyama, Itoyo Tokuda, Kaori Sawada, Kazushige Ihara.

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
