## [Decision Letter · Decision Letter 0]

2 Jan 2020

PONE-D-19-29066

Association between higher urinary normetanephrine and insulin resistance in a general population

PLOS ONE

Dear Prof. Daimon,

Thank you for submitting your manuscript to PLOS ONE. After careful consideration, we feel that it has merit but does not fully meet PLOS ONE’s publication criteria as it currently stands. Therefore, we invite you to submit a revised version of the manuscript that addresses the points raised during the review process.

We would appreciate receiving your revised manuscript by Feb 16 2020 11:59PM. To enhance the reproducibility of your results, we recommend that if applicable you deposit your laboratory protocols in protocols.io, where a protocol can be assigned its own identifier (DOI) such that it can be cited independently in the future. For instructions see: http://journals.plos.org/plosone/s/submission-guidelines#loc-laboratory-protocols

We look forward to receiving your revised manuscript.

Kind regards,

Tatsuo Shimosawa, M.D., Ph.D.

Academic Editor

PLOS ONE

Journal Requirements:

2. Please refer to any post-hoc corrections to correct for multiple comparisons during your statistical analyses. If these were not performed please justify the reasons.

Please refer to our statistical reporting guidelines for assistance (https://journals.plos.org/plosone/s/submission-guidelines.#loc-statistical-reporting).

Reviewers' comments:

Reviewer's Responses to Questions

**Comments to the Author**

1. Is the manuscript technically sound, and do the data support the conclusions?

Reviewer #1: Yes

Reviewer #2: Yes

2. Has the statistical analysis been performed appropriately and rigorously? 

Reviewer #1: Yes

Reviewer #2: No

3. Have the authors made all data underlying the findings in their manuscript fully available?

Reviewer #1: Yes

Reviewer #2: Yes

4. Is the manuscript presented in an intelligible fashion and written in standard English?

Reviewer #1: Yes

Reviewer #2: Yes

5. Review Comments to the Author

Reviewer #1: This article is very interesting because authors investigated association between urinary metanephrine/normetanephrine and insulin resistance/secretion in a large cohort. I think this study has enough novelty and priority. Almost all of investigations and analyses were appropriate and performed adequately, but I would like to suggest authors to confirm some points.

1. Authors quoted various accurate references, particularly about the relationship between catecholamines and glycemic characteristics. However, I would like you to check one sentence, which was very important. Authors of ref. 17 revealed differences in the actions of adrenaline and noradrenaline with regard to glucose intolerance in patients with pheochromocytoma. In the article, authors of ref. 17 described “Regression analysis revealed that the improvement in HOMA-B from before to after surgery had a significant positive association with the improvement in urinary levels of metanephrine after surgery (P = 0.0286), and a significant negative association with the improvement in urinary levels of normetanephrine after surgery (P = 0.0248). The improvement in HOMA-IR did not show a positive association with the improvement in urinary levels of metanephrine but showed a significant positive association with the improvement in urinary levels of normetanephrine (P = 0.0001)” in Results of ref. 17. However, authors wrote “the improvement in U-M concentrations was "negatively" associated with the improvement in HOMA-B” (sentence 77-79). “the improvement in urinary metanephrine was “positively” associated with the improvement in HOMA-B” must be proper. In discussion, authors’ description was proper (sentence 229-232), so I think authors miswrote. Authors should check ref. 17 again, and then modify this description.

2. As for description, authors must be more careful. Authors should modify below, and then check whole manuscript once more.

Sentence 31 “normetanephrine (U-NM) and metanephrine (U-M) “should be modified to “urinary normetanephrine (U-NM) and metanephrine (U-M)”.

Sentence 44 “HOMA-B” is incorrect, I think. Authors should modify this to “HOMA-R”.

Sentence 67 “throughα 1” should be modified to “through α1”.

Sentence 73 “normetanephrin” should be modified to “normetanephrine”.

Sentence 76 “changes in U-NM and U-MN concentrations” should be modified to “changes in U-M and U-NM concentrations”.

Sentence 212, 215, 219 Authors wrote “CA concentrations”, “CA concentration”, and “CAs concentrations” respectively. Authors should unify the writing.

Reviewer #2: I enjoyed reading this manuscript, in particular, the detailed analyses using appropriate statistical models, and the explanation of the results based on the analyses. However, I have some concerns.

(a) NO sample size/power statements are provided, and that needs to be justified wrt. the sample size of the analysis.

(b) Results on logistic regression should be reported in terms of effects sizes, p-values AND an indication of the uncertainty (typically, 95% confidence intervals, henceforth, CIs), given that p-values and CIs provide complementary information; see article https://www.ncbi.nlm.nih.gov/pmc/articles/PMC2689604/

Furthermore, results should be explained that way.

(c) The title of the paper says "General population"; however, the target population is Japanese. This can be confusing to readers!, given that the authors are not consider a global population (spanning across all continents). The title should be revised.

6. PLOS authors have the option to publish the peer review history of their article (what does this mean?). If published, this will include your full peer review and any attached files.

Reviewer #1: No

Reviewer #2: No

---

## [Author Response · Author response to Decision Letter 0]

13 Jan 2020

Dear reviewer #1:

 Thank you very much for having reviewed our manuscript entitled " Association between higher urinary normetanephrine and insulin resistance in a general population: PONE-D-19-29066". We have changed our manuscript to fulfill your criticisms as much as possible, and in most cases, I just changed as you requested. In any case, we beg your generosity to kindly feel satisfied with my responses. The details of the responses to each criticism were written below. 

 This article is very interesting because authors investigated association between urinary metanephrine/normetanephrine and insulin resistance/secretion in a large cohort. I think this study has enough novelty and priority. Almost all of investigations and analyses were appropriate and performed adequately, but I would like to suggest authors to confirm some points.

-----I appreciate your evaluation.

1. Authors quoted various accurate references, particularly about the relationship between catecholamines and glycemic characteristics. However, I would like you to check one sentence, which was very important. Authors of ref. 17 revealed differences in the actions of adrenaline and noradrenaline with regard to glucose intolerance in patients with pheochromocytoma. In the article, authors of ref. 17 described “Regression analysis revealed that the improvement in HOMA-B from before to after surgery had a significant positive association with the improvement in urinary levels of metanephrine after surgery (P = 0.0286), and a significant negative association with the improvement in urinary levels of normetanephrine after surgery (P = 0.0248). The improvement in HOMA-IR did not show a positive association with the improvement in urinary levels of metanephrine but showed a significant positive association with the improvement in urinary levels of normetanephrine (P = 0.0001)” in Results of ref. 17. However, authors wrote “the improvement in U-M concentrations was "negatively" associated with the improvement in HOMA-B” (sentence 77-79). “the improvement in urinary metanephrine was “positively” associated with the improvement in HOMA-B” must be proper. In discussion, authors’ description was proper (sentence 229-232), so I think authors miswrote. Authors should check ref. 17 again, and then modify this description.

-----Thanks for your pointing. We correct the miswriting as you pointed.

2. As for description, authors must be more careful. Authors should modify below, and then check whole manuscript once more.

-----Thanks for your comments and I apologize such miswriting. We corrected as you suggested as follows.

Sentence 31 “normetanephrine (U-NM) and metanephrine (U-M) “should be modified to “urinary normetanephrine (U-NM) and metanephrine (U-M)”.

------Thanks. We corrected them as you mentioned.

Sentence 44 “HOMA-B” is incorrect, I think. Authors should modify this to “HOMA-R”.

------ Thanks. We corrected them as you mentioned.

Sentence 67 “throughα 1” should be modified to “through α1”.

------- Thanks. We corrected them as you mentioned.

Sentence 73 “normetanephrin” should be modified to “normetanephrine”.

------ Thanks. We corrected them as you mentioned.

Sentence 76 “changes in U-NM and U-MN concentrations” should be modified to “changes in U-M and U-NM concentrations”.

----- Thanks. We corrected them as you mentioned.

Sentence 212, 215, 219 Authors wrote “CA concentrations”, “CA concentration”, and “CAs concentrations” respectively. Authors should unify the writing.

------ Thanks. We unified the words as “CA concentrations”. 

Dear reviewer #2:

 Thank you very much for having reviewed our manuscript entitled " Association between higher urinary normetanephrine and insulin resistance in a general population: PONE-D-19-29066". We have changed our manuscript to fulfill your criticisms as much as possible, and in most cases, I just changed as you requested. In any case, we beg your generosity to kindly feel satisfied with my responses. The details of the responses to each criticism were written below. 

I enjoyed reading this manuscript, in particular, the detailed analyses using appropriate statistical models, and the explanation of the results based on the analyses. However, I have some concerns.

-----Thanks for your evaluation.

(a) NO sample size/power statements are provided, and that needs to be justified wrt. the sample size of the analysis.

-----Thanks for your comment. We added such statement in “discussion” as follows: Contrary, adjustment for multiple factors can …….statistical power………..was 99.8% to detect an OR of 2.73 for insulin resistance at a significance level of 0.05, ………….

(b) Results on logistic regression should be reported in terms of effects sizes, p-values AND an indication of the uncertainty (typically, 95% confidence intervals, henceforth, CIs), given that p-values and CIs provide complementary information; see article https://www.ncbi.nlm.nih.gov/pmc/articles/PMC2689604/

Furthermore, results should be explained that way.

-----Thanks for your comments. We added p values for the analyses in “Results”. 

(c) The title of the paper says "General population"; however, the target population is Japanese. This can be confusing to readers!, given that the authors are not consider a global population (spanning across all continents). The title should be revised.

-----Thanks for your comments. We corrected the title as follows:…in a Japanese population.

---

## [Decision Letter · Decision Letter 1]

24 Jan 2020

Association between higher urinary normetanephrine and insulin resistance in a Japanese population

PONE-D-19-29066R1

Dear Dr. Daimon,

We are pleased to inform you that your manuscript has been judged scientifically suitable for publication and will be formally accepted for publication once it complies with all outstanding technical requirements.

With kind regards,

Tatsuo Shimosawa, M.D., Ph.D.

Academic Editor

PLOS ONE

Additional Editor Comments (optional):

Reviewers' comments:

Reviewer's Responses to Questions

**Comments to the Author**

1. If the authors have adequately addressed your comments raised in a previous round of review and you feel that this manuscript is now acceptable for publication, you may indicate that here to bypass the “Comments to the Author” section, enter your conflict of interest statement in the “Confidential to Editor” section, and submit your "Accept" recommendation.

Reviewer #1: All comments have been addressed

Reviewer #2: All comments have been addressed

2. Is the manuscript technically sound, and do the data support the conclusions?

Reviewer #1: Yes

Reviewer #2: Yes

3. Has the statistical analysis been performed appropriately and rigorously? 

Reviewer #1: Yes

Reviewer #2: Yes

4. Have the authors made all data underlying the findings in their manuscript fully available?

Reviewer #1: Yes

Reviewer #2: No

5. Is the manuscript presented in an intelligible fashion and written in standard English?

Reviewer #1: Yes

Reviewer #2: Yes

6. Review Comments to the Author

Reviewer #1: Thank you for addressing my concerns. The revised manuscript is suitable for publication. There were no points to be modify.

Reviewer #2: The authors addressed my comments from the previous round adequately. I have no further comments. Can the authors provide some suggestions on how the real data can be accessed?

7. PLOS authors have the option to publish the peer review history of their article (what does this mean?). If published, this will include your full peer review and any attached files.

Reviewer #1: No

Reviewer #2: No

---

## [Editor Report · Acceptance letter]

29 Jan 2020

PONE-D-19-29066R1 

Association between higher urinary normetanephrine and insulin resistance in a Japanese population 

Dear Dr. Daimon:

I am pleased to inform you that your manuscript has been deemed suitable for publication in PLOS ONE. Congratulations! Your manuscript is now with our production department. 

With kind regards,

on behalf of

Prof. Tatsuo Shimosawa 

Academic Editor

PLOS ONE